**Data Availability Statement:** The data used in this study are third-party data from the Korean Nation Health and Nutrition Examination Survey (https://

# Gender-based comparison of factors affecting regular exercise of patients with Non-Insulin Dependent Diabetes Mellitus (NIDDM) based on the 7th Korea National Health and Nutrition Examination Survey (KNHANES)

**Ji-Yeon Choi[1]ʘ, Jieun Shin[2]ʘ *, Seunghui Baek[3]***

**1** Department of Nursing, Gimcheon University, Gimcheon City, Gyungbuk, Korea, **2** Department of Biomedical Informatics, College of Medicine, Konyang University, Daejeon, Korea, **3** Department of Health Exercise Management, Sungshin Women's University, Seoul, Korea

ʘ These authors contributed equally to this work.
* jeshin@konyang.ac.kr (JS); sh100@sungshin.ac.kr (SB)

## Abstract

### Objective

This study aimed to determine the gender factors that influence regular exercise in patients with non-insulin-dependent diabetes mellitus (NIDDM) in Korea.

### Methods

A total of 1,432 patients with NIDDM were recruited using raw data from the Korea National Health and Nutrition Survey conducted between 2016 and 2018. SAS 9.4 was adopted for data analyses, and the distributional difference was measured with multinomial logistic regression and Rao-Scott x2 statistics to identify the factors that influence the regular physical activities of patients. that the analysis only provides associations.

### Results

Based on general characteristics, health behaviors, and conditions, patients with NIDDM in Korea were less physically active. In addition, patients with higher educational attainment, higher income, and higher subjective health conditions had a higher odds ratio for regular exercise. Meanwhile, the ratio was lower for smokers and those stressed up.

### Conclusion

A professional guide for the initial phase of training and consistent management is required to increase the involvement of patients with NIDDM in regular exercise. Therefore, it is important to maintain their motivation to continue exercising. Rather than providing a universal guideline, it is more important to provide customized programs and management plans

knhanes.kdca.go.kr/knhanes/sub03/sub03_02_05.
do) and can be accessed following the protocol
outlined in the Methods section.

**Funding:** The authors received no specific funding
for this work.

**Competing interests:** The authors have declared
that no competing interests exist.

which reflect factors that influence their engagement in physical activities, such as individual physical strength, stress level, alcohol consumption, and arthritis.

# Introduction

## Background

The number of people suffering from diabetes mellitus (DM) worldwide increased from approximately 320 million in 2014 to 425 million in 2017. In 2017, 4 million people lost their lives to DM and related complications. This trend implies that the prevalence rate of DM is likely to further increase, with 693 million estimated patients in 2045 [1].

The Republic of Korea (hereinafter "Korea") ranks 5th in the mortality rate of patients with DM among the Organization for Economic Cooperation and Development countries. Such an increase in the prevalence rate of DM may be related to complications and relevant death rates. According to the statistical report on the causes of death announced by Korea in 2018, DM was the 6th disease resulting in mortality [2]. DM is classified into type 1 (insulin-dependent diabetes mellitus [IDDM]) and type 2 (non-insulin-dependent diabetes mellitus [NIDDM]) depending on the cause. IDDM occurs when the beta cells in the pancreas are destroyed, causing an absolute shortage of insulin. This type requires insulin as treatment. NIDDM accounts for approximately 90% of all patients with DM in Korea. Causes vary from family history to stress, obesity, lack of exercise, intake of excessive energy, and environmental conditions [1, 3].

The causes of NIDDM differ, but regardless of the cause and type, all patients with DM should receive appropriate treatment to prevent complications. Possible complications include chronic renal failure, cardiovascular disease, blindness, and hypoesthesia, which worsen the quality of life [3].

The American Diabetes Association (ADA) emphasized that the absence of self-management after developing NIDDM causes complications that eventually lead to death. The medical staff should, therefore, seek various preventable factors that affect the condition and suggest to the patient an appropriate method of managing the body condition [4].

Carrying out physical activities is essential for the management of NIDDM. They should be performed regardless of the type of disease; regular exercise reduces the required amount of insulin and reduces insulin resistance in IDDM, while preventing or delaying the occurrence of NIDDM. Exercise also prevents complications in the heart and circulatory system. When a patient realizes that the purpose of treating NIDDM is not to reduce the level of blood sugar but to prevent all sorts of possible complications from occurring in the circulatory system, the importance of exercising would not be emphasized more in patients with NIDDM. In addition, conditions of high blood pressure, dyslipidemia, and obesity, which are other risk factors for coronary artery disease, can be improved [5]. Previous studies have reported that physical activities improve metabolism, which is critical for improving glycated hemoglobin, blood sugar, and insulin sensitivity status [6]. The guideline announced by the Korean Diabetes Association recommends that patients perform moderately intense exercise with 50%–70% of the maximal heart rate for > 150 min and aerobic exercise for at least three times a week to manage blood sugar, maintain body weight, and reduce risks of developing cardiovascular diseases [5]. However, exercising regularly for a long period of time is challenging. According to previous research on the exercising habit of patients with NIDDM in Korea, about 45.7% of the patients were not involved in physical activities at all in a week, and the rate of exercise

three times a week, as recommended by the Korea Diabetes Association, was 45.1%, which falls far below 69% of their counterparts in the US [7]. Therefore, regular exercise and methods to increase physical activities in daily life are highly recommended for patients with NIDDM. Depending on the lifestyle pattern of each patient, methods to maximize physical activities along with regular workout routines are emphasized as one of the core values of self-management [8].

The pros and cons of an adequate amount of physical activity affect both males and females [9], but the level of such activities is reported to be different depending on gender. In a study conducted on the elderly with a chronic disease, males tended to engage in physical activities that promote health more frequently than females [10], and in another group, males walked more often than their female counterparts [11]. In addition, in previous studies, males and females showed a difference in physiological, socio-structural, and psychological factors [10, 12], and in health behaviors such as smoking, alcohol consumption, and dietary control [12, 13]. In addition, males and females showed different results in terms of health conditions, such as physical functions, chronic diseases, stress cognition, and pain [12]. Both genders were found to have different physiological, socio-structural, and psychological factors [9, 12], and their interest in practicing health-related activities has been on the rise in various preceding studies. Terminologies such as health-keeping habits, health practices, self-nursing, and healthy lifestyle were mainly used when referring to health behaviors, with measurements based on exercise routines, dietary control, smoking, alcohol consumption, and safety [14]. Therefore, it is necessary to identify how the characteristic differences between men and women affect the physical activities of patients with NIDDM.

Research has been conducted on the overall health behaviors of patients with NIDDM in Korea, but as they provide broad information about regular workouts [14], it is limited to identifying characteristics unique to individuals or a gender that may affect their exercising activity. Hence, it is considered that analyzing predictors of regular exercise depending on the gender of patients with NIDDM will provide important primary data for planning differentiated strategies to promote the physical activities of each sex. Therefore, this study aimed to provide basic data for developing programs that support physical activities of each sex by examining factors that affect exercise habits in patients with NIDDM based on their general gender characteristics, health behaviors, and conditions with raw data from the 7th Korea National Health and Nutrition Examination Survey (KNHANES).

## Purpose

This study aimed to provide information necessary for planning intervention strategies to promote the exercise of patients with NIDDM by examining and comparing relevant elements that affect their workout habits (both males and females) based on the raw data from the 7th Korea National Health and Nutrition Examination Survey (KNHANES) conducted between 2016 and 2017.

The objectives are as follows:

- To compare regular workout routines of patients with NIDDM based on general gender characteristics.

- To compare regular workout routines of patients with NIDDM based on the health behaviors of each gender.

- To compare regular workout routines of patients with NIDDM based on the physical conditions of each gender.

- To identify elements that affect regular workouts of both male and female patients with NIDDM.

## Materials and methods

### Design

This is a secondary analysis study conducted with the approval to use the raw data of the 7th KNHANES (2016–2017) executed by the Korea Disease Control and Prevention Agency (KDCA; https://knhanes.kdca.go.kr) This descriptive research aims to identify the differences between male and female patients with NIDDM depending on general characteristics, health behaviors, and health conditions for carrying out regular exercise, and to analyze the related affective elements.

### Research variables

**Diabetes mellitus.** Diabetes mellitus (DM) is a disease that occurs when insulin secretion fails or when the level of insulin resistance increases within peripheral tissues. DM is classified into type 1 (insulin-dependent diabetes mellitus[IDDM]), type 2 (non-insulin-dependent diabetes mellitus: NIDDM), and others. A patient is diagnosed with NIDDM when the level of glycated hemoglobin is $\geq$ 6.5%, 8-hour fasting plasma glucose of $\geq$ 126 mg/dL, and 2-hour plasma blood sugar of $\geq$ 200 mg/dL, along with typical symptoms (polyuria, polydipsia, and polyphagia). The key count used in the diagnosis, management, and prediction of complications in NIDDM today is glycated hemoglobin [1].

**Regular workout.** Regarding regular workouts, patients who spent $\geq$ 2.5 hours doing medium-level physical activities, $\geq$ 1.25 hours of intensive physical activities, or mixed the two (1 min for intensive and 2 min for medium-level) were classified as carrying out regular workouts, and others were classified as not engaged in regular exercise. In addition, walking for $>$ 10 min (going to school/work, traveling, or exercising) was also included as a regular workout [15].

**Health behavior.** Smoking, alcohol consumption, obesity, changes in body weight, and dietary habits were used to identify health behaviors. For smoking, the "current smoking rate" was adopted, which is the number of current smokers who have smoked $>$ 100 cigarettes (5 packs) in their lifetime divided by the number of patients $>$ 19 years of age. Raw data that classified smoking in the past with non-smoking as "no" and currently smoking as "yes" were used. For alcohol consumption, "monthly drinking rate" was used, which is the number of people who responded that they had consumed alcohol at least once a month for the recent year divided by the number of patients $>$ 19 years of age. The analysis was based on the raw data that classified those who had $<$ 1 glass of alcohol a month during the past year or non-drinkers as "no" and those who have $>$ 1 glass a month during the same period as "yes." In regards to the degree of obesity, Body Mass Index (BMI) was used, which is a calculation of body mass using height and weight. The raw data were based on the following figures: normal (BMI $\geq$ 18.5 kg/m$^2$, $<$ 25 kg/m$^2$) and obese (BMI $\geq$ 25 kg/m$^2$) [16]. For changes in body weight, raw data that have classified patients into "no change," "lose weight," and "gained weight" depending on their response to the question: "Did you experience any change in body weight compared to last year?" was used. Regarding dietary control, raw data where patients responded with "Yes" and "No" to the question asking whether they are adhering to a dietary method was used.

**Health condition.** Comorbid diseases, subjective health conditions, limitations in activity, and arthritis were referred for analyzing factors of health condition. For comorbid diseases,

patients with at least one of the following diseases were considered and classified as having comorbidities: blood pressure, stroke, asthma, NIDDM, thyroid disease, kidney failure, chronic obstructive pulmonary disease, dyslipidemia, pulmonary tuberculosis, hepatitis B, hepatitis C, and cirrhosis. In regards to the subjective health condition, based on the responses to the question: "What do you think about your health these days?" Patients were reclassified into "very good," "good," "okay," "bad," and "very bad." Meanwhile, for limitations in activities, the raw data that classified patients who responded to the question: "Are you experiencing limitations in carrying out daily life or social activities due to any health conditions or physical/mental disorders?" into "Yes" and "No" was used. For stress cognition, the responses of the patients to the question "How much do you feel stressed in daily life?" were reclassified ("very much" and "very much" into "yes," "little bit" and "seldom" into "no"). For arthritis, the raw data that classified those who currently have the condition as "yes" and others as "no" was utilized.

## Patients and analysis

Among 6,583 patients with NIDDM (Weighted N = 13,283,291) who participated in the 7th KNHANES, 3,314 were males and 1,159 were females. Each number could be generalized as 7,384,230 and 5,899,060 based on the complex sample design.

Components of the complex sample design, such as stratification, cluster sampling, and weight, were used, and data analysis utilized the complex sampling procedure of the SPSS/WIN 25 program. To integrate the yearly data from the 7th KNHANES, the proportion of samples from each year (2016, 2017) was multiplied by the existing weight to calculate the integrated weight. In addition, the statistical analysis was based on the Rao-Scott x2. To identify factors influencing the regular workout of patients, significant variables analyzed from the general characteristics, health behaviors, and health conditions were used as independent variables for multiple logistic regression analysis. The odds ratio (OR) and confidence interval (95% confidence interval, 95% [CI]) were used to describe the statistics.

## Results

### Characteristics of the patients

The results of the analysis of the characteristics of the study participants are shown in Table 1. Regarding general characteristics, 68.6% of the participants were engaged in physical activities. Males and females accounted for 71.5% and 64.8%, respectively (p < .001). In terms of age, among the individuals aged 40 or above, males between 50 and 60 years accounted for the greatest proportion (32.8%), while females ≥ 70 accounted for the largest proportion (30.6%). Regarding educational status, 36.5% of males were college graduates, which was the largest proportion, while for females, the largest proportion was high-school graduates (31.9%, p < .001). In terms of occupation, 76.8% of males had a job, while only 44.9% of females were economically active, showing a significant difference (p < .001). Regarding income status, 65.5% of males had a source of income, while it was only 50.5% for females, also showing a significant difference (p < .001).

A significant difference was also observed in health behaviors, where 34.6% of males were smokers, while only 4.6% of females were smokers (p < .001). Males also showed a higher percentage of alcohol consumption; 93.9% of males drank alcohol, while 74.4% for females (p < .001). In terms of obesity, 48.7% of men and 46.2% of women were obese (p>.05), and for changes in body weight, 70.8% and 63.1% of men and women showed no difference, respectively (p < .001).

**Table 1. Characteristics of patients.**

| | | Male (N = 7,384,230) | | Female (N = 5,899,060) | | Total | | $\chi^2$ | p |
|---|---|---|---|---|---|---|---|---|---|
| Age | <50 | 709 | (27.4) | 485 | (18.0) | 1194 | (23.2) | 219.941 | < .0001 |
| | <60 | 846 | (32.8) | 702 | (26.1) | 1548 | (29.8) | | |
| | <70 | 880 | (23.7) | 923 | (25.2) | 1803 | (24.4) | | |
| | > = 70 | 879 | (16.0) | 1159 | (30.6) | 2038 | (22.5) | | |
| Spouse | No | 461 | (14.1) | 1125 | (32.8) | 1586 | (22.4) | 271.457 | < .0001 |
| | Yes | 2851 | (85.9) | 2141 | (67.2) | 4992 | (77.6) | | |
| Education | Elementary | 648 | (15.4) | 1379 | (40.2) | 2027 | (26.5) | 486.934 | < .0001 |
| | Middle | 463 | (14.4) | 444 | (13.7) | 907 | (14.0) | | |
| | High | 982 | (33.7) | 781 | (29.6) | 1763 | (31.9) | | |
| | College | 988 | (36.5) | 451 | (16.5) | 1439 | (27.6) | | |
| Job | No | 923 | (23.2) | 1732 | (55.1) | 2655 | (37.4) | 495.665 | < .0001 |
| | Yes | 2159 | (76.8) | 1324 | (44.9) | 3483 | (62.6) | | |
| Income | No | 1361 | (34.5) | 1753 | (49.5) | 3114 | (41.1) | 139.488 | < .0001 |
| | Yes | 1936 | (65.5) | 1498 | (50.5) | 3434 | (58.9) | | |
| Smoking | No | 2258 | (65.4) | 3125 | (95.4) | 5383 | (78.7) | 711.404 | < .0001 |
| | Yes | 1052 | (34.6) | 143 | (4.6) | 1195 | (21.3) | | |
| Alcohol | No | 226 | (6.1) | 906 | (25.6) | 1132 | (14.8) | 398.451 | < .0001 |
| | Yes | 3084 | (93.9) | 2362 | (74.4) | 5446 | (85.2) | | |
| BMI | Normal | 1759 | (51.3) | 1755 | (53.8) | 3514 | (52.4) | 2.804 | 0.094 |
| | Obese | 1541 | (48.7) | 1490 | (46.2) | 3031 | (47.6) | | |
| Changes in body weight | No change | 2338 | (70.8) | 2076 | (63.1) | 4414 | (67.4) | 46.731 | < .0001 |
| | Decreased | 448 | (13.8) | 428 | (13.5) | 876 | (13.7) | | |
| | Increased | 465 | (15.4) | 688 | (23.4) | 1153 | (18.9) | | |
| Comorbidity | No | 2148 | (68.8) | 1863 | (60.3) | 4011 | (65.0) | 44.686 | < .0001 |
| | Yes | 1162 | (31.2) | 1401 | (39.7) | 2563 | (35.0) | | |
| Subjective body condition | Good | 2435 | (80.3) | 2123 | (70.4) | 4558 | (75.9) | 55.247 | < .0001 |
| | Bad | 676 | (19.7) | 967 | (29.6) | 1643 | (24.1) | | |
| Stress | Low | 2551 | (76.9) | 2385 | (75.3) | 4936 | (76.2) | 1.832 | 0.176 |
| | High | 697 | (23.1) | 805 | (24.7) | 1502 | (23.8) | | |
| Activity | No | 2770 | (90.9) | 2622 | (86.5) | 5392 | (88.9) | 21.601 | < .0001 |
| | Yes | 331 | (9.1) | 459 | (13.5) | 790 | (11.1) | | |
| Arthritis | No | 2867 | (94.3) | 2133 | (72.3) | 5000 | (84.5) | 446.198 | < .0001 |
| | Yes | 241 | (5.7) | 954 | (27.7) | 1195 | (15.5) | | |
| Regular Workout | No | 733 | (28.5) | 848 | (35.2) | 1581 | (31.4) | 19.671 | < .0001 |
| | Yes | 1729 | (71.5) | 1480 | (64.8) | 3209 | (68.6) | | |

Regarding health conditions, 31.2% of males and 39.7% of females had comorbidities, showing that it occurred more frequently in females (p < .001). For subjective health conditions, 80.3% of males and 70.4% of females replied that they felt their body was in a good state (p < .001). The two genders did not show a significant difference in terms of stress cognition; 23.1% of males and 24.7% of females had a "high" level of stress (p>.05). Females showed a higher percentage of 13.5% for limitations in activity than their male counterparts (9.1%; p < .001). They also had arthritis than males; 27.7% of females had joint inflammation, while only 5.7% of males had the condition (p < .001).

### Differences in carrying out regular physical activities based on general characteristics

The difference between male and female patients with NIDDM in carrying out regular physical activities based on general characteristics is shown in Table 2.

Overall, patients with NIDDM showed statistically significant differences depending on age, the existence of husband/wife, and educational and income status. Those > 70 years were least engaged in physical activities, while those who had their spouse showed a high level of engagement. In addition, patients with a high educational level and good income were more likely to participate in workouts.

Regarding gender, male patients showed a statistically significant difference in exercise depending on their educational and income status. The higher the two figures, the higher the exercise rate.

For female patients, a statistically significant difference appeared depending on their age, existence of their spouse, and educational and income status. Those above > 70 years were least engaged in physical activities, while just as their male counterparts, those who had their husbands were more willing to exercise. The result was the same for education and income levels, where the higher figure shown in the two categories led to a higher participation rate in workouts.

### Differences in regular workout habits based on health behaviors

The differences between male and female patients with NIDDM in conducting regular exercise based on health behaviors are presented in Table 3.

From the overall perspective, including both male and female patients, they showed a statistically significant difference in smoking and alcohol consumption. Non-smokers were more engaged in physical activities, but non-drinkers were less engaged in such activities.

Both male and female patients showed a statistically significant difference in smoking, where smokers were found participating more in physical activities.

**Table 2. Differences in regular workout habits based on general characteristics.**

| | | Total | | | | | | Male | | | | | | Female | | | | | |
|---|---|---|---|---|---|---|---|---|---|---|---|---|---|---|---|---|---|---|---|
| | | No | | Yes | | $\chi^2$ | p | No | | Yes | | $\chi^2$ | p | No | | Yes | | $\chi^2$ | p |
| Age | <50 | 305 | (30.0) | 628 | (70.0) | 21.825 | < .0001 | 173 | (28.4) | 382 | (71.6) | 2.612 | 0.455 | 132 | (33.2) | 246 | (66.8) | 26.660 | < .0001 |
| | <60 | 380 | (31.3) | 816 | (68.7) | | | 195 | (30.0) | 437 | (70.0) | | | 185 | (33.1) | 379 | (66.9) | | |
| | <70 | 395 | (27.7) | 986 | (72.3) | | | 169 | (25.8) | 500 | (74.2) | | | 226 | (30.0) | 486 | (70.0) | | |
| | > = 70 | 501 | (38.3) | 779 | (61.7) | | | 196 | (29.7) | 410 | (70.3) | | | 305 | (45.0) | 369 | (55.0) | | |
| Spouse | No | 396 | (36.7) | 643 | (63.3) | 11.071 | 0.001 | 119 | (32.9) | 222 | (67.1) | 3.036 | 0.082 | 277 | (39.0) | 421 | (61.0) | 4.690 | 0.030 |
| | Yes | 1185 | (30.1) | 2564 | (69.9) | | | 614 | (27.8) | 1507 | (72.2) | | | 571 | (33.7) | 1057 | (66.3) | | |
| Education | Elementary | 560 | (40.3) | 785 | (59.7) | 51.014 | < .0001 | 170 | (35.8) | 272 | (64.2) | 19.422 | 0.000 | 390 | (42.5) | 513 | (57.5) | 24.121 | < .0001 |
| | Middle | 226 | (33.0) | 468 | (67.0) | | | 107 | (32.6) | 233 | (67.4) | | | 119 | (33.3) | 235 | (66.7) | | |
| | High | 443 | (30.0) | 1011 | (70.0) | | | 238 | (30.3) | 551 | (69.7) | | | 205 | (29.7) | 460 | (70.3) | | |
| | College | 347 | (25.8) | 937 | (74.2) | | | 216 | (23.4) | 669 | (76.6) | | | 131 | (32.5) | 268 | (67.5) | | |
| Job | No | 693 | (32.3) | 1346 | (67.7) | 0.968 | 0.325 | 219 | (27.9) | 523 | (72.1) | 0.107 | 0.744 | 474 | (34.8) | 823 | (65.2) | 0.121 | 0.728 |
| | Yes | 884 | (30.8) | 1855 | (69.2) | | | 512 | (28.7) | 1202 | (71.3) | | | 372 | (35.6) | 653 | (64.4) | | |
| Income | No | 802 | (37.8) | 1294 | (62.2) | 46.844 | < .0001 | 339 | (35.4) | 611 | (64.6) | 22.480 | < .0001 | 463 | (40.0) | 683 | (60.0) | 17.140 | < .0001 |
| | Yes | 771 | (27.5) | 1906 | (72.5) | | | 392 | (25.3) | 1111 | (74.7) | | | 379 | (31.0) | 795 | (69.0) | | |

**Table 3. Differences in regular workout habits based on health behaviors.**

| | | Total | | | | | Male | | | | | Female | | | | |
|---|---|---|---|---|---|---|---|---|---|---|---|---|---|---|---|---|
| | | No | | Yes | | $\chi^2$ | p | No | | Yes | | $\chi^2$ | p | No | | Yes | | $\chi^2$ | p |
| Smoking | No | 1265 | (30.5) | 2671 | (69.5) | 4.359 | 0.037 | 463 | (26.1) | 1243 | (73.9) | 9.960 | 0.002 | 802 | (34.6) | 1428 | (65.4) | 4.296 | 0.038 |
| | Yes | 316 | (34.7) | 538 | (65.3) | | | 270 | (33.4) | 486 | (66.6) | | | 46 | (47.0) | 52 | (53.0) | | |
| Alcohol Consumption | No | 249 | (36.2) | 411 | (63.8) | 5.517 | 0.019 | 40 | (31.6) | 85 | (68.4) | 0.442 | 0.506 | 209 | (37.4) | 326 | (62.6) | 0.914 | 0.339 |
| | Yes | 1332 | (30.8) | 2798 | (69.2) | | | 693 | (28.4) | 1644 | (71.6) | | | 639 | (34.6) | 1154 | (65.4) | | |
| BMI (kg/m$^2$) | Normal | 840 | (31.2) | 1715 | (68.8) | 0.041 | 0.840 | 393 | (29.0) | 894 | (71.0) | 0.256 | 0.613 | 447 | (33.9) | 821 | (66.1) | 1.372 | 0.241 |
| | Obese | 734 | (31.6) | 1488 | (68.4) | | | 338 | (28.0) | 831 | (72.0) | | | 396 | (36.6) | 657 | (63.4) | | |
| Changes in Weight | No Change | 1069 | (31.2) | 2197 | (68.8) | 0.227 | 0.893 | 518 | (28.2) | 1260 | (71.8) | 0.617 | 0.735 | 551 | (35.6) | 937 | (64.4) | 0.820 | 0.664 |
| | Reduced | 220 | (31.6) | 397 | (68.4) | | | 109 | (28.1) | 219 | (71.9) | | | 111 | (36.6) | 178 | (63.4) | | |
| | Gained | 290 | (32.1) | 607 | (67.9) | | | 106 | (30.5) | 246 | (69.5) | | | 184 | (33.4) | 361 | (66.6) | | |

## Differences in regular workout habits based on health conditions

The differences between male and female patients with NIDDM participating in physical activities based on health conditions are presented in Table 4.

From a comprehensive perspective, including both genders, a statistically significant difference was shown in terms of subjective perception of health conditions, stress level, and limitations in activity. Those who had a better perception of their health conditions were more engaged in physical activities, while those with less stress and without limitations in activity tended to exercise more.

For male patients with NIDDM, a statistically significant difference was observed in the subjective perception of their body condition, where the greater figure led to a higher participation rate in physical activities.

Meanwhile, female patients with NIDDM appeared to have a statistically significant difference in terms of subjective perception of their body condition, stress level, and limitations in activity. Those who thought they were in a good condition, had less stress, and were not limited in carrying out activities were more willing to participate in physical activities.

## Factors influencing regular exercise and the result in patients with NIDDM

Multinomial logistic regression analysis was conducted to identify the factors influencing regular exercise in patients with NIDDM, and the results are shown in Table 5.

**Table 4. Differences in regular workout habits based on physical conditions.**

| | | Total | | | | | | Male | | | | | | Female | | | | |
|---|---|---|---|---|---|---|---|---|---|---|---|---|---|---|---|---|---|---|---|
| | | No | | Yes | | $\chi^2$ | p | No | | Yes | | $\chi^2$ | p | No | | Yes | | $\chi^2$ | p |
| Comorbidity | No | 932 | (30.4) | 1998 | (69.6) | 3.214 | 0.073 | 460 | (27.6) | 1131 | (72.4) | 1.614 | 0.204 | 472 | (34.6) | 867 | (65.4) | 0.416 | 0.519 |
| | Yes | 649 | (33.2) | 1211 | (66.8) | | | 273 | (30.4) | 598 | (69.6) | | | 376 | (36.1) | 613 | (63.9) | | |
| Subjective Body Condition | Good | 1128 | (29.4) | 2542 | (70.6) | 27.224 | < .0001 | 551 | (26.7) | 1441 | (73.3) | 17.739 | < .0001 | 577 | (33.4) | 1101 | (66.6) | 7.729 | 0.005 |
| | Bad | 453 | (39.0) | 666 | (61.0) | | | 182 | (37.6) | 287 | (62.4) | | | 271 | (40.2) | 379 | (59.8) | | |
| mh_stress | Low | 1180 | (30.1) | 2544 | (69.9) | 8.590 | 0.003 | 565 | (27.8) | 1393 | (72.2) | 1.946 | 0.163 | 615 | (33.3) | 1151 | (66.7) | 8.875 | 0.003 |
| | High | 399 | (36.0) | 657 | (64.0) | | | 168 | (31.4) | 332 | (68.6) | | | 231 | (41.4) | 325 | (58.6) | | |
| Activity | No | 1385 | (30.8) | 2906 | (69.2) | 8.600 | 0.003 | 654 | (28.0) | 1587 | (72.0) | 2.890 | 0.089 | 731 | (34.5) | 1319 | (65.5) | 4.137 | 0.042 |
| | Yes | 195 | (37.9) | 303 | (62.1) | | | 78 | (34.4) | 142 | (65.6) | | | 117 | (41.2) | 161 | (58.8) | | |
| Arthritis | No | 1274 | (31.0) | 2674 | (69.0) | 2.723 | 0.099 | 676 | (28.6) | 1606 | (71.4) | 0.331 | 0.565 | 598 | (34.9) | 1068 | (65.1) | 0.321 | 0.571 |
| | Yes | 307 | (34.1) | 535 | (65.9) | | | 57 | (26.5) | 123 | (73.5) | | | 250 | (36.2) | 412 | (63.8) | | |

mh stress: When they responded that they feel a lot of stress during their daily lives, stressfulness.

**Table 5. Analysis of affective factors for regular exercise.**

| | | All | Male | Female |
|---|---|---|---|---|
| | | OR (95% CI) | OR (95% CI) | OR (95% CI) |
| Age | ≤ 59 | 1.02 (0.82–1.26) | 1.00 (0.75–1.33) | 1.02 (0.75–1.40) |
| | ≤ 69 | 1.47 (1.14–1.89) | 1.43 (1.03–1.99) | 1.40 (0.96–2.05) |
| | ≥ 70 | 1.09 (0.83–1.43) | 1.39 (0.96–2.01) | 0.80 (0.52–1.23) |
| Spouse | Yes | 1.01 (0.83–1.23) | 0.92 (0.68–1.25) | 0.92 (0.72–1.17) |
| Education | Middle school | 1.27 (1.01–1.59) | 1.16 (0.81–1.68) | 1.28 (0.93–1.76) |
| | High school | 1.45 (1.15–1.83) | 1.31 (0.94–1.83) | 1.49 (1.08–2.05) |
| | College | 1.77 (1.39–2.24) | 1.76 (1.24–2.52) | 1.32 (0.93–1.87) |
| Income | Yes | 1.35 (1.14–1.61) | 1.48 (1.14–1.91) | 1.27 (0.99–1.60) |
| Smoking | Yes | 0.80 (0.66–0.97) | 0.76 (0.60–0.97) | 0.59 (0.36–0.98) |
| Alcohol Consumption | Yes | 1.12 (0.90–1.40) | 1.20 (0.74–1.94) | 0.98 (0.74–1.29) |
| Subjective Body Condition | Good | 1.33 (1.10–1.60) | 1.48 (1.13–1.93) | 1.16 (0.91–1.47) |
| Stress | Yes | 0.81 (0.68–0.98) | 0.90 (0.69–1.17) | 0.74 (0.58–0.95) |
| Activity | Yes | 1.02 (0.80–1.30) | 1.08 (0.71–1.63) | 0.94 (0.69–1.29) |

For the univariate analysis, the patients showed a statistically significant difference in factors including age, existence of a spouse, educational level, income status, smoking, alcohol consumption, subjective perception of body conditions, stress level, and limitations in activity. Female patients with NIDDM had a statistically significant difference in all variables except drinking, while male patients showed a statistically significant difference in educational level, income status, smoking, and subjective health conditions. This study adopted variables that showed a significant difference among the overall patients with NIDDM as independent variables.

Among all the patients with NIDDM, the odds ratio for the "regularly exercising group" was found to be 1.47 times higher (95% CI 1.14–1.89) among those > 60 years compared to those in their 40s. Also, the difference in the ratio increased with better educational status; compared to elementary school graduates, middle-school graduates had a 1.27 (1.01–1.59) times higher ratio, 1.45 (1.15–1.83) times higher for high-school graduates, and 1.77 (1.39–2.24) times higher for college graduates. Also, the odds ratio for engaging in workouts was 1.35 (1.14–1.61) times higher for those with income, while the figure for smokers was 0.8 (0.66–0.97) times lower. The ratio was 1.33 (1.10–1.60) times higher for people with "good" subjective physical conditions, and patients with "high" stress levels had a 0.81 (0.68–0.98) times lower odds ratio for exercising.

The odds ratio of male patients with NIDDM classified as a "regular exerciser" was 1.76-(1.24–2.52) times higher for college graduates than for elementary graduates, and those with a source of income showed a 1.48 (1.14–1.91) times higher ratio. The figure was 0.76 (0.60–0.97) times lower for smokers. Also, patients with "good" subjective physical conditions were appeared to have a 1.48 (1.13–1.93) times higher odds ratio of carrying out physical activities.

Meanwhile, the odds ratio of female patients with NIDDM aged > 60 years classified as a "regular exerciser" was twice (95% CI 1.24–3.21) as higher than those in their 40s. In terms of educational level, high-school graduates had a 1.49 (1.08–2.05) times higher ratio than elementary school graduates. In addition, highly stressed patients had a 0.74 (0.58–0.95) times lower odds ratio.

## Discussion

An adequate level of physical activity for patients with NIDDM is an efficient way to delay the deterioration of health conditions and lower the expenses of clinical trials [17–19]. Therefore,

this study was conducted to provide basic data for developing programs that encourage physical activities by identifying the general characteristics of patients with NIDDM (both male and female), their exercise rate, and relevant elements that affect such behavior.

An evaluation the extent to which patients with NIDDM were engaged in physical activities revealed that only 68.6% of the patients went on walks regularly, where males (71.5%) were found to be more willing to exercise than females (64.8%) by 6.7%. This result was similar to the conclusion of the Global Physical Activity Questionnaire (GPAQ) that 41.5% of patients with NIDDM carried out aerobic exercise on a regular basis and that male patients were more physically active than their female counterparts [20]. In the United States, 31% of patients with NIDDM are not engaged in a regular workout routine, and 38% are even below the recommended level of physical activity [21]. Meanwhile, about 60% of Portuguese patients with NIDDM do not perform any type of exercise [22]. Patients with NIDDM in Korea tend to participate more in physical activities on a regular basis; however, it was found that patients with NIDDM across the world seldom exercise [17–19].

Comparing the engagement of patients with NIDDM in regular workouts based on general characteristics, patients showed a significant difference (p < .05) depending on age, educational level, and income status. This result is in line with previous studies that show how patients with NIDDM who exercise regularly differ significantly depending on age, sex, income status, and educational level [20]. Regarding the age group of both sexes, the exercise rate was high in the 60s, 40s, and 70s, but it was the lowest at 61.7% for those over 70 years. Male patients did not show a significant difference in terms of age, while their female counterparts showed a significant difference (p < .05). Patients with NIDDM walked more frequently than the average walking ratio of adults in Korea in 2018 (34.1% for those in their 50s, 41.6% in their 60s, and 32.1% in those in their 70s) [23]. However, the two results were similar in that the figures dropped sharply for those ≥ 70. Walking generally requires muscular strength in the lower body, but the fact that the human body drastically loses approximately 25% of its physical strength in the lower body with age; from about 70 years, [24] seems to have led to a drop in the walking rate.

In this study, the factors influencing regular exercise based on healthy behaviors were significantly different depending on BMI level and alcohol consumption. The gender analysis showed that males with normal body weight were significantly more engaged in physical activities than those who were obese, while women who consume alcohol showed a higher tendency to exercise.

These results were contradictory to those of a study where among 1,480 patients with NIDDM [21], there was no difference in physical activity level and BMI level. However, regarding the walking rate of patients with NIDDM based on the BMI level, the result is in line with a study showing that the normal group (male: 31%, female: 25%) had a higher exercise rate than the obese group (21%, 22.7%) for both males and females [25] Another study showed that adult males were more willing to walk in the order of normal weight, overweight, obese, and low weight as they observed the walking rate of adult males for ≥ 4 days [26]. Weight loss through moderate physical activity lowered the risk of NIDDM by up to 58% [27], and a weight loss of 1 kg reduced the risk of NIDDM by 16%. Therefore, increased physical activity is very important [28]. In addition, according to the recent announcement of the Irish Society for Clinical Nutrition & Metabolism, NIDDM conditions can improve by reducing only 15% of body weight in patients with NIDDM [29]. This highlights the importance of losing weight by participating more in physical activities.

For patients' engagement in regular exercise depending on alcohol consumption, the exercise rate was significantly higher in females when they consumed more than one glass of alcohol per month. This result is similar to a study that showed that people who drink more than

one glass of alcohol a month had a high rate of walking regularly [11], and if they had a drinking habit, the rate of carrying out physical activities to promote health was high [10]. In addition, there are studies that show that the incidence rates of NIDDM and coronary artery disease were lower by 33%–56% and 55%, respectively [30] when consuming three glasses of alcohol per day compared to other cases. However, some studies have shown that drinking negatively affects physical activity [10]. Therefore, further studies are required to prove the relationship between alcohol consumption and regular exercise in patients with NIDDM.

Meanwhile, a significant difference was observed for factors such as subjective health condition, limitations in activity, and arthritis in the number of patients with NIDDM participating in physical activities on a regular basis based on health conditions. In addition, while both males and females showed a significant difference in subjective health conditions, females also showed such differences in limitations in activity and arthritis.

Moreover, regarding the exercise rate analysis based on the subjective health status, both males and females who answered that their health status is "good" did not perform exercise than those who answered "not good." In other words, they seemed to exercise more, if they thought they were not healthy. These results were also reported in a study of elderly individuals with chronic diseases. Both elderly males and females were less engaged in physical activity when they perceived their body condition to be bad [10]. In another study, subjective health condition was a major factor in performing exercise on a regular basis [11]. In addition, the exercise rate was lower for males when they were more stressed up. These results are similar to other studies in which people who exercised regularly had lower stress than those who stopped exercising or who did not exercise at all [31]. The results of a study revealed that aerobic exercise provides physiological and psychological resistance to stress [32]; therefore, regular exercise seems to be a stress-relieving factor.

Females were less engaged in physical activities if they had limitations in activity or arthritis. This is similar to the result of another study that showed that females who have low back pain were twice as less exercising than others, and that elderly suffering from chronic pain experience irritation in daily life due to pain and fatigue [33]. In addition, in the 12-month exercise intervention study on cardiac rehabilitation, the dropout rate of females was higher than that of males. This was found to be a musculoskeletal problem. In the case of females, more often than males, musculoskeletal problems and cardiovascular complications coexist, and the occurrence of fractures due to falls or accidents is the cause of dropout in exercise programs [34]. Therefore, detailed attention is required for female patients because such factors have a greater influence on females. Pain is a factor that reduces physical activity, but physical activity above the recommended level positively affects pain, fatigue, and daily life [33]. Therefore, women with limitations in activity and arthritis need to be encouraged to exercise. However, women who are relatively more vulnerable to musculoskeletal problems than men will need exercise prescriptions, taking this part into account.

In this study, regarding the factors influencing the regular exercise of patients with NIDDM, all patients with NIDDM and females in their "60s" had a 1.47 times higher probability of regular exercise than those in their "40s." Compared to "elementary school graduates," the exercise rate was higher in the order of "middle school graduates and high school graduates." In addition, if the patients' subjective health condition was "Yes (good)," the exercise rate was higher. For males, the odds ratio to be classified as carrying out regular exercise was 1.48 higher when the response was "Yes (good)" for the subjective health condition. In other words, it was found that the exercise rate of young patients with NIDDM in their 40s was low along with the level of education. According to previous studies, the reason for the low rate of exercise among patients with NIDDM was reported to be a lack of awareness and specific knowledge about how beneficial workouts can be [35–37].

NIDDM is a disease that lasts for a lifetime and, if not managed, causes serious complications. Hence, continued instruction and support are required [38]. However, improving lifestyle habits such as dietary control and having an adequate amount of workout decreases the risk of NIDDM [39], especially since moderate exercise rapidly improves blood sugar and insulin action [40]. Therefore, regular physical activity is a very important factor for patients with NIDDM. Recently, ADA recommended that people with NIDDM should avoid sedentary activities. If the time spent sitting is reduced by 2.5 h a day, energy consumption can be increased by 350 kcal to prevent weight gain [41]. In addition, walking for 1 min after meals is effective in maintaining blood sugar [42]. As such, a mild increase in physical activity is very helpful in improving the condition of patients with NIDDM. Hence, it is important to emphasize the importance of self-management through physical activities and education related to exercise for recently diagnosed patients with NIDDM.

## Conclusion

Analysis of the factors that influence the physical activities of patients with NIDDM based on their general characteristics, health behaviors, and health status of patients with NIDDM showed that patients with NIDDM in Korea were less engaged in such activities. A low regular exercise rate was found in patients who were young, had lower educational levels, or were in poor physical condition.

Therefore, efforts such as initial exercise guidance and continuous exercise management should be made to increase the regular exercise rate of patients with NIDDM. Above all, it is important to maintain motivation to exercise, and it is believed that it is more important to provide and manage customized exercise programs that reflect factors that affect patients' engagement in exercises, such as individual physical fitness, stress, alcohol, and arthritis, rather than universal exercise guidelines.

This study is meaningful in that it analyzes the affective factors for the regular exercise habits of patients with NIDDM in Korea, and provides basic data for the development of exercise promotion programs considering gender. In addition, the results of the study can be generalized as they are based on a large sample of patients with NIDDM in Korea. However, there is a limitation in that the causal relationship is unclear because of the nature of the cross-sectional study. In addition, there is a limit that does not include factors that affect individual behavior, such as attitudes, perceptions, and values. Hence, a large-scale study that encompasses such factors should be conducted in the future.

## Author Contributions

**Conceptualization:** Ji-Yeon Choi, Seunghui Baek.

**Data curation:** Jieun Shin.

**Formal analysis:** Jieun Shin.

**Methodology:** Ji-Yeon Choi, Jieun Shin, Seunghui Baek.

**Supervision:** Seunghui Baek.

**Visualization:** Jieun Shin.

**Writing – original draft:** Ji-Yeon Choi, Jieun Shin, Seunghui Baek.

**Writing – review & editing:** Ji-Yeon Choi, Jieun Shin, Seunghui Baek.

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
