## [Decision Letter · Decision Letter 0]

26 Apr 2021

PONE-D-21-05083

Gender-Based Comparison of Factors Affecting Regular Exercise of Diabetic Patients : Based on the 7th Korea National Health and Nutrition Examination Survey (KNHANES)

PLOS ONE

Dear Dr. seunghui,

Thank you for submitting your manuscript to PLOS ONE. After careful consideration, we feel that it has merit but does not fully meet PLOS ONE’s publication criteria as it currently stands. Therefore, we invite you to submit a revised version of the manuscript that addresses the points raised during the review process.

We look forward to receiving your revised manuscript.

Kind regards,

Antonio Palazón-Bru, PhD

Academic Editor

PLOS ONE

Journal Requirements:

Reviewers' comments:

Reviewer's Responses to Questions

**Comments to the Author**

1. Is the manuscript technically sound, and do the data support the conclusions?

Reviewer #1: Yes

2. Has the statistical analysis been performed appropriately and rigorously? 

Reviewer #1: I Don't Know

3. Have the authors made all data underlying the findings in their manuscript fully available?

Reviewer #1: Yes

4. Is the manuscript presented in an intelligible fashion and written in standard English?

Reviewer #1: No

5. Review Comments to the Author

Reviewer #1: The authors investigate data from the KNHANES study with the aim to understand the gender factors that influence the regular exercise of patients with diabetes in Korea.

This kind of investigation is highly relevant for cardiovascular medicine, enabling personalized prevention, which is considered to dampen disease trajectories and thereby reduce healthcare costs in the future while increasing health span in the general population, but also in people with increased cardiovascular risk such as patients with diabetes. We know National investigations of this kind are important to better address local differences in lifestyle (and potentially genetic influences). Data from National studies are usually accepted better by policy makers and authorities. The study provides a wealth of results that can hopefully be used to implement prevention programmes.

The variables studied are clearly described.

The study does not discriminate between type 1 and type 2 diabetes, albeit both are different disease entities. This needs tob e amended.

In the abstract, phrasing needs tob e revised in order to better reflect that the analysis only provides associations, not causalities.

Minor:

There are several grammatical mistakes and misleading phrases in the text. I would suggest to ask a native English speaker to proof-read the manuscript.

line 77-78: now: „… slightly intense exercise with the maximum heart rate of 50-70% for more than 150 minutes…“ suggest to change to „…moderately intense exercise with 50-70% of the maximal heart rate for more than 150 minutes…“

line 89: now: „…males tended to be engaged in physical activities that promote health than females…“ suggest to change into „…males tended to be engaged in physical activities that promote health more frequently than females…“

line 90: „…males were walking regularly than their female counterparts…“ suggest to change into „…males were walking regularly more often than their female counterparts…“

Table 4: Please explain the abbreviation „mh_stress“

6. PLOS authors have the option to publish the peer review history of their article (what does this mean?). If published, this will include your full peer review and any attached files.

Reviewer #1: No

---

## [Author Response · Author response to Decision Letter 0]

20 Aug 2021

Reviewers' comments: Reviewer's Responses to Questions

Comments to the Author

1. Is the manuscript technically sound, and do the data support the conclusions?

Reviewer #1: Yes

2. Has the statistical analysis been performed appropriately and rigorously?

Reviewer #1: I Don't Know

Response: Our research was conducted in accordance with the analysis guidelines of the data provider (the Korea Centers for Disease Control and Prevention (KCDC)) and analyzed by statisticians.

3. Have the authors made all data underlying the findings in their manuscript fully available?

Reviewer #1: Yes

4. Is the manuscript presented in an intelligible fashion and written in standard English?

Reviewer #1: No

Response: We requested a specialized institution to edit English. ________________________________________

5. Review Comments to the Author

Reviewer #1: The authors investigate data from the KNHANES study with the aim to understand the gender factors that influence the regular exercise of patients with diabetes in Korea.

This kind of investigation is highly relevant for cardiovascular medicine, enabling personalized prevention, which is considered to dampen disease trajectories and thereby reduce healthcare costs in the future while increasing health span in the general population, but also in people with increased cardiovascular risk such as patients with diabetes. We know National investigations of this kind are important to better address local differences in lifestyle (and potentially genetic influences). Data from National studies are usually accepted better by policy makers and authorities. The study provides a wealth of results that can hopefully be used to implement prevention programmes.

The variables studied are clearly described.

The study does not discriminate between type 1 and type 2 diabetes, albeit both are different disease entities. This needs tob e amended.

Response: Thanks for pointing out. Diabetes was changed to Diabetes mellitus (DM), Type 1 to Insulin Dependent Diabetes Mellitus (IDDM), and Type 2 to Non-insulin Dependent Diabetes Mellitus (NIDDM).

In the abstract, phrasing needs tob e revised in order to better reflect that the analysis only provides associations, not causalities.

Response: that the analysis only provides associations.

Minor:

There are several grammatical mistakes and misleading phrases in the text. I would suggest to ask a native English speaker to proof-read the manuscript.

line 77-78: now: „… slightly intense exercise with the maximum heart rate of 50-70% for more than 150 minutes…“ suggest to change to „…moderately intense exercise with 50-70% of the maximal heart rate for more than 150 minutes…“

line 89: now: „…males tended to be engaged in physical activities that promote health than females…“ suggest to change into „…males tended to be engaged in physical activities that promote health more frequently than females…“

line 90: „…males were walking regularly than their female counterparts…“ suggest to change into „…males were walking regularly more often than their female counterparts…“

Response : We have corrected all of the above. (The color is displayed differently.)

Table 4: Please explain the abbreviation „mh_stress“

Response : “mh stress: When they responded that they feel a lot of stress during their daily lives, stressfulness” I added it as a sentence below the table

Thank you So much for checking in detail.________________________________________

6. PLOS authors have the option to publish the peer review history of their article (what does this mean?). If published, this will include your full peer review and any attached files.

Do you want your identity to be public for this peer review? For information about this choice, including consent withdrawal, please see our Privacy Policy.

Reviewer #1: No

---

## [Decision Letter · Decision Letter 1]

13 Sep 2021

Gender-Based Comparison of Factors Affecting Regular Exercise of Diabetic Patients : Based on the 7th Korea National Health and Nutrition Examination Survey (KNHANES)

PONE-D-21-05083R1

Dear Dr. seunghui,

We’re pleased to inform you that your manuscript has been judged scientifically suitable for publication and will be formally accepted for publication once it meets all outstanding technical requirements.

Kind regards,

Antonio Palazón-Bru, PhD

Academic Editor

PLOS ONE

Additional Editor Comments (optional):

Reviewers' comments:

Reviewer's Responses to Questions

**Comments to the Author**

1. If the authors have adequately addressed your comments raised in a previous round of review and you feel that this manuscript is now acceptable for publication, you may indicate that here to bypass the “Comments to the Author” section, enter your conflict of interest statement in the “Confidential to Editor” section, and submit your "Accept" recommendation.

Reviewer #1: All comments have been addressed

2. Is the manuscript technically sound, and do the data support the conclusions?

Reviewer #1: Yes

3. Has the statistical analysis been performed appropriately and rigorously? 

Reviewer #1: Yes

4. Have the authors made all data underlying the findings in their manuscript fully available?

Reviewer #1: Yes

5. Is the manuscript presented in an intelligible fashion and written in standard English?

Reviewer #1: Yes

6. Review Comments to the Author

Reviewer #1: All of my previous questions have been adressed. I have no further comments.

xxxxxxxxxxxxxxxxxxxxxxxx

7. PLOS authors have the option to publish the peer review history of their article (what does this mean?). If published, this will include your full peer review and any attached files.

Reviewer #1: No

---

## [Editor Report · Acceptance letter]

23 Sep 2021

PONE-D-21-05083R1 

Gender-based comparison of factors affecting regular exercise of patients with Non- Insulin Dependent Diabetes Mellitus (NIDDM) based on the 7th Korea National Health and Nutrition Examination Survey (KNHANES) 

Dear Dr. Baek:

I'm pleased to inform you that your manuscript has been deemed suitable for publication in PLOS ONE. Congratulations! Your manuscript is now with our production department. 

Kind regards, 

on behalf of

Dr. Antonio Palazón-Bru 

Academic Editor

PLOS ONE